# Contribution of Hypoalbuminemia and Anemia to the Prognostic Value of Plasma p-Cresyl Sulfate and p-Cresyl Glucuronide for Cardiovascular Outcome in Chronic Kidney Disease

**DOI:** 10.3390/jpm12081239

**Published:** 2022-07-28

**Authors:** Francis Verbeke, Raymond Vanholder, Wim Van Biesen, Griet Glorieux

**Affiliations:** Department of Internal Medicine and Pediatrics, Nephrology Section, Ghent University Hospital, 9000 Ghent, Belgium; francis.verbeke@uzgent.be (F.V.); raymond.vanholder@ugent.be (R.V.); wim.vanbiesen@ugent.be (W.V.B.)

**Keywords:** chronic kidney disease, uremic toxins, p-cresyl sulfate, p-cresyl glucuronide, cardiovascular outcome, albumin, hemoglobin

## Abstract

Free plasma concentrations of protein-bound uremic toxins (PBUTs) may be influenced by serum albumin and hemoglobin. The potential association of serum albumin and hemoglobin with free levels of p-cresyl sulfate (pCS) and p-cresyl glucuronide (pCG) and their predictive value for cardiovascular morbidity and mortality were explored. A total of 523 non-dialysis chronic kidney disease (CKD) stages G1–G5 patients were prospectively followed for the occurrence of fatal or non-fatal cardiovascular events over a 5.5-year period. A negative correlation was found between albumin and between hemoglobin, and both total and free pCS and pCG. In multiple linear regression, PBUTs were negatively associated with eGFR (estimated glomerular filtration rate) and hemoglobin but not albumin. In multivariate Cox regression analysis, albumin was a predictor of outcome, independent of pCS and pCG, without interactions between albumin and pCS or pCG. The relation of low hemoglobin with adverse outcome was lost when albumin was entered into the model. Lower concentrations of pCS and pCG are associated with higher serum albumin and hemoglobin. This may indicate that there are two pathways in the blood that potentially contribute to attenuating the vasculotoxic effects of these PBUTs. The association of PBUTs with cardiovascular risk is not explained by albumin levels, which remains a strong and independent predictor for adverse outcome.

## 1. Introduction

Among the large number of uremic retention solutes that accumulate with chronic kidney disease (CKD), protein-bound uremic toxins (PBUTs) are highly ranked for biochemical, pathophysiological and clinical evidence of toxicity [1]. These compounds have been associated with oxidative stress and endothelial activation in vitro and in animal studies, and have been repeatedly associated with cardiovascular (CV) and overall mortality in epidemiological studies of patients with chronic kidney disease (CKD) [1,2]. We recently reported an association between total and free concentrations of p-cresyl sulfate (pCS) and p-cresyl glucuronide (pCG) and major cardiovascular events [3]. In that study, the strongest association was found for free pCS in multivariate-adjusted models with statistical correction for multiple comparisons. As for many drugs, it could be assumed that the biological effects of these toxins relate to their circulating free concentrations and thus to the degree of protein binding, mainly to albumin, in the plasma. The erythrocytes are another important constituent of the blood. Our group previously demonstrated in a series of in vitro experiments that PBUTs are also distributed within erythrocytes by active transport mechanisms involving Band 3 proteins, anion exchangers located on erythrocyte membranes [4].

The aim of the present study was to further explore the potential association of serum albumin and hemoglobin concentrations with free levels of pCS and pCG, and their predictive value for cardiovascular morbidity and mortality.

## 2. Materials and Methods

### 2.1. Study Population

The methods have been described previously [3]. In brief, a panel of uremic toxins was measured in a cohort of 523 patients with CKD stages G1–G5 not on dialysis, which was then followed for 5.5 years. In the present analysis, the primary endpoint was the composite of non-fatal and fatal cardiovascular events (CVEs). CVEs were defined as acute myocardial infarction, acute coronary syndrome, cerebrovascular accident, transient ischemic attack, coronary artery bypass grafting, percutaneous transluminal coronary angioplasty, peripheral revascularization, other CVEs, heart failure and sudden death. This study was approved by the local ethics committee of Ghent University Hospital, and written informed consent was obtained from all participants before sample collection.

In the present analysis, we focus on pCS and pCG since these were most consistently related to cardiovascular events and mortality in our previous analysis [3].

### 2.2. Laboratory Analysis

An aliquot of frozen serum was thawed for the batch quantification of albumin and prealbumin at the clinical laboratory of the Ghent University Hospital in Belgium. Albumin and prealbumin concentrations were determined by nephelometry on a Behring Nephelometer analyzer II (Siemens Medical Solutions, Erlangen, Germany) according to published standards [5]. Reference values for albumin were 35–52 g/L. Reference values for hemoglobin were 13.5–17.0 g/dL for men and 11.7–15.1 g/dL for women.

### 2.3. Statistical Analysis

Associations between continuous variables were studied according to Spearman rank correlations. Independent associations between variables that may affect free and total concentrations of the PBUTs were explored using multiple linear regression. Concentrations of PBUTs were log transformed. To investigate the potential prognostic role of PBUTs, albumin, prealbumin and hemoglobin, Cox proportional hazards models were fitted; hazard ratios (HRs), 95% confidence intervals (CIs) and statistical significance were estimated from these models. To assess the potential effect of serum albumin on the prognostic role of PBUTs, an interaction was tested. Hazard ratios are expressed per 1 standard deviation (SD) for continuous variables. The assumption of proportionality of hazards was checked using log (–log(survival)) plots. The models were adjusted for age and sex, systolic blood pressure and diabetes mellitus. All tests were 2-sided and a *p* value <0.05 was considered significant. All analyses were performed using IBM^®^ SPSS^®^ Statistics for windows version 25 (IBM, Armonk, NY, USA).

## 3. Results

Population characteristics have been published previously [3]. In brief, the patients included had a median age of 66 (51–76) years and 42% were female. A total of 12% of the patients were in CKD G1, 16% in CKD G2, 49% in CKD G3, 19% in CKD G4 and 3% in CKD G5.

### 3.1. Correlation Studies

In the present analysis, we found a weak but significant negative correlation between serum albumin and total (T) and free (F) plasma concentrations of pCS and pCG. A similar negative correlation was also found for hemoglobin (Table 1).

No significant correlations were found between prealbumin and the respective PBUTs. Concentrations of PBUTs according to the quartiles of serum albumin and hemoglobin are shown in Figure 1.

In a multiple linear regression model using eGFR as a covariate, concentrations of all PBUT concentration variables were negatively associated with eGFR and hemoglobin but not with albumin (model for FpCS shown in Table 2, models for TpCS, TpCG and FpCG were similar; see Appendix A).

### 3.2. Outcome

In multivariate models adjusted for age, sex, systolic blood pressure and diabetes mellitus, total and free concentrations of pCS and pCG and albumin were significantly associated with the primary outcome of fatal or non-fatal CVEs, whereas prealbumin was not. Only a trend for a slightly higher risk associated with lower body mass index (BMI) was observed (model with FpCS, HR 0.97, 95%CI 0.94–1.05, *p* = 0.096).

Hemoglobin was also associated with outcome, but its effect was lost when albumin was entered into the model. The final model for log (FpCS) is shown in Table 3. Models were similar for TpCS, TpCG and FpCG (see Appendix A).

## 4. Discussion

The present study shows that free as well as total concentrations of pCS and pCG are inversely correlated with albumin, but not with prealbumin in patients with CKD not on dialysis. In addition, the latter PBUT concentrations are also inversely related with hemoglobin levels, independent of kidney function. However, despite the significant correlation between albumin and plasma levels of PBUTs, both remain strong and independent predictors of cardiovascular morbidity and mortality in multivariate analysis. For instance, one standard deviation in log-transformed free pCS is associated with an increased risk that is comparable with a 20 mmHg increase in systolic blood pressure. For albumin, a 3.5 g/L increase is associated with an approximately 30% lower risk for CVEs.

Our data are in line with a small number of studies on the prognostic value of PBUTs that have also included albumin in their multivariate analysis. Barreto et al. analyzed indoxyl sulfate (IS) in a mixed group of 139 CKD patients, also including patients on dialysis, and found that IS remained independently associated with all-cause mortality after multivariate adjustment including serum albumin [6]. In another study, albumin was related to the free fraction of pCS but was not independently associated with all-cause mortality, perhaps because its effect was already captured by the Davies comorbidity score and subjective global assessment in that model [7]. In a subgroup analysis of the HEMO, study levels of total pCS and total IS were only associated with a higher risk of cardiac death among patients with a serum albumin <36 g/L [8]. This suggests that albumin modulates the toxic effect of PBUT, perhaps by an effect on the free fraction. Unfortunately, free fractions were not reliable in the HEMO study because blood samples were collected after heparin administration, which may have led to artificially increased free solute concentrations [9,10]. The present study extends the above findings to patients with CKD not on dialysis and showed that the predictive value of free concentrations is not mediated by an effect of albumin. This may have several reasons.

The number of patients with hypoalbuminemia in our study may have been too low to detect a potential modulating effect on the predictive effect of free concentrations of PBUTs. Indeed, the median albumin concentration in the lowest quartile was still 39.5 g/L, with only 10 patients recording values below 35 g/L.

Free and total concentrations of PBUTs may also be affected by transport into erythrocytes. Indeed, our group previously demonstrated that PBUTs are distributed within erythrocytes, with the fastest rates of transport observed for indole acetic acid (IAA) and for pCS [4]. In the present analyses, free and total concentrations of pCS and pCG were inversely related with hemoglobin, an effect that was independent of eGFR as a potential confounder, as higher concentrations of PBUT and lower hemoglobin could both be caused by lower GFR. A univariate inverse association has also been previously reported in a smaller group of 139 patients with CKD, also including patients on hemodialysis, but the association was lost when eGFR was included in the model [11]. In our study, hemoglobin is also a predictor of outcome, but this effect was lost after the inclusion of albumin in the multivariate model. It could be that both binding to albumin as well as transport into erythrocytes represent two complementary mechanisms that try to defend the vascular endothelium against the toxic effect of circulating uremic toxins.

Finally, albumin may be linked to adverse outcome by mechanisms that are independent of the effects of uremic toxins. Indeed, serum albumin has been linked to the risk of all-cause and cardiovascular mortality in several populations, including middle-aged men, patients who had undergone transcatheter aortic valve replacement, the general population [12,13], elderly persons [14] as well as in patients receiving dialysis [15,16,17,18,19]. The mechanism by which serum albumin potentially contributes to cardiovascular disease and mortality is likely multifactorial.

Albumin may be a surrogate for nutritional condition and frailty which is associated with adverse outcome [12]. Nevertheless, in the present study, prealbumin—also used as a nutrition marker—was not related to outcome, whereas a trend was only observed for BMI. The role of prealbumin as a nutrition marker has been debated [20]. Prealbumin, also known as transthyretin, has a plasma half-life of approximately 2 days, much shorter than that of albumin which has a relatively large body pool and a half-life of 20 days. Prealbumin reflects recent dietary intake rather than overall nutritional status [21], which may be better reflected by albumin. Our data thus indicate that in order to assess the impact of nutritional status on hard endpoints, routine anthropometric and laboratory tests are poor surrogates for nutritional status. Consequently, the collection of dietary information is necessary. This information is also required for studying and devising dietary interventions with the aim of decreasing the plasma levels of PBUTs which originate from the intestinal digestion of dietary products, followed by proteolytic fermentation by gut bacteria.

Albumin also affects coagulation by exerting a heparin-like effect that is likely due to binding antithrombin, which enhances the neutralization of coagulation factor Xa [22], and by inhibiting platelet activation [23]. It is a major determinant of oncotic pressure, and low albumin may increase blood viscosity and increase thrombotic risk [24]. Albumin has a favorable effect on the endothelial function. In vitro and animal models have shown that albumin inhibits the TNFα-induced upregulation of VCAM-1 expression and monocyte adhesion to the endothelial wall, and improves the bioavailability of nitric oxide by protecting its bioactivity and increasing its half-life in vivo [25]. Albumin contains thiol groups which can help scavenge reactive oxygen and nitrogen species, hence decreasing oxidative stress [25].

Albumin is also a negative acute phase reactant, with decreased production in the setting of inflammation [26]. In our study, somewhat surprisingly, the levels of high sensitivity C-reactive protein were not related with albumin or with outcome. This may be due to the fact that only patients in stable clinical condition, without infectious or inflammatory conditions, were included. Another reason may be that a large proportion of patients were treated with statins which have known anti-inflammatory effects and favorably affect cardiovascular outcome.

Our study has several strengths. We consistently included albumin, prealbumin and hemoglobin into the same multivariate models, exploring their potential modulating effect on the association between concentrations of PBUT and outcome, which resulted in consistent models for both pCS and pCG. Our study covered the whole range of CKD stages G1–G5 patients and excluded dialyzed patients, thus reducing the heterogeneity of the studied population. Patients were prospectively followed over a sufficient amount of time to allow the detection of relevant endpoints. Ultimately, follow-up was completed for 99% of patients, and all endpoints were identified by the same physician based on clinical reality (i.e., not based on administrative codes using International Classification of Diseases codes), therefore avoiding misclassification of outcomes. The limitations encountered in the study were the underrepresentation of non-Caucasian ethnicities, the lack of repeated measurements and the absence of adjustment for potential residual confounding factors. In addition, we possessed no information on nutritional status as assessed through food questionnaires.

## 5. Conclusions

Concentrations of albumin and hemoglobin, the two major constituents of the blood, correlate with free and total concentrations of the PBUTs pCS and pCG in patients with CKD not on dialysis. They may represent two pathways in the human blood that potentially contribute to attenuating the vasculotoxic effects of these UTs as they accumulate with progressive decline in kidney function. The association of PBUTs with cardiovascular risk is not explained by albumin binding, which remains a strong and independent predictor for adverse outcome. Future studies should explore whether nutritional interventions can improve albumin concentrations and eventually outcome, using food questionnaires as a tool to document nutritional status. Studies on erythropoietin-stimulating agents could provide an opportunity to further explore the role of PBUT transportation into erythrocytes in vascular toxicity and the outcome.

## Figures and Tables

**Figure 1 jpm-12-01239-f001:**
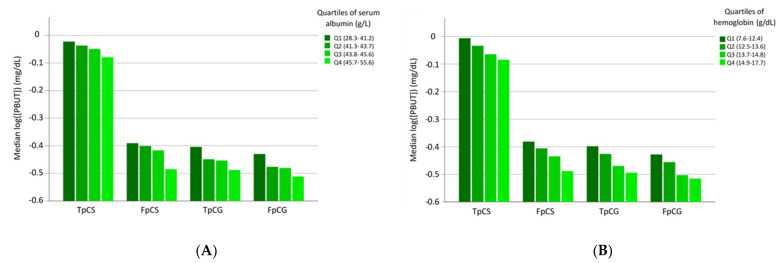
Concentrations of uremic toxins according to quartiles of serum albumin (panel (**A**)) and hemoglobin (panel (**B**)). TpCS: total p-cresyl sulfate; FpCS: free p-cresyl sulfate; TpCG: total p-cresyl glucuronide; FpCG: free p-cresyl glucuronide (pCG). Albumin correlated with prealbumin (*r* = 0.33), eGFR (*r* = 0.20) and hemoglobin (*r* = 0.41) (all *p* < 0.001).

**Table 1 jpm-12-01239-t001:** Correlation between albumin, hemoglobin and protein-bound uremic toxins.

PBUT	Correlation Coefficients	
	Albumin	Hemoglobin
TpCS	−0.157	−0.284
FpCS	−0.250	−0.319
TpCG	−0.163	−0.271
FpCG	−0.192	−0.270

PBUT: protein-bound uremic toxin; T: total fraction; F: free (unbound) fraction; pCS: p-cresyl sulfate; pCG: p-cresyl glucuronide. Spearman rank correlation, all *p*-values < 0.001.

**Table 2 jpm-12-01239-t002:** Regression model of factors associated with free pCS concentrations.

Factor	Beta Coefficient	95% CI	*p*
(Constant)		−2.297; −0.488	0.003
Hemoglobin (g/dL)	−0.149	−1.133; −0.040	<0.001
Albumin (g/L)	−0.059	−0.039; 0.005	0.14
eGFR (mL/min/1.73 m^2^)	−0.501	−0.021; −0.016	<0.001

CI: confidence interval; eGFR: estimated glomerular filtration rate.

**Table 3 jpm-12-01239-t003:** Association between log (FpCS) and outcome.

Variable	HR	95% CI	*p*
Age (1 SD)	1.70	1.34; 2.18	<0.001
Sex (male = 1)	1.81	1.27; 2.57	0.001
Diabetes (yes = 1)	2.65	1.89; 3.71	<0.001
SBP (1 SD)	1.36	1.16; 1.60	<0.001
Log(FpCS) (1 SD)	1.37	1.14; 1.65	0.001
Albumin (1 SD)	0.63	0.53; 0.75	<0.001

Hazard ratio’s (HR) for continuous variables are expressed per 1 standard deviation (SD) (age = 17 years, SBP = 20 mmHg, albumin = 3.46 g/L); CI: confidence interval; FpCS: free p-cresyl sulfate; SBP: systolic blood pressure.

## Data Availability

All data are fully available without restriction, in an anonymized format, upon request.

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
