# Peer review of "Contribution of Hypoalbuminemia and Anemia to the Prognostic Value of Plasma p-Cresyl Sulfate and p-Cresyl Glucuronide for Cardiovascular Outcome in Chronic Kidney Disease"

_jpm, 2022, doi:10.3390/jpm12081239_

Round 1

Reviewer 1 Report

The authors used the data part from their previous study. They tried to disclose that the factors of hypoalbuminemia and anemia might affect the effect of PCS and PCG on CV outcome. However,  the statistical methods used in this manuscript didn't sufficiently support its conclusion. Therefore, there were some major flaws in the manuscript.

The major comments

1.     The idea should be more precise. Some factors can affect the effect of PCS and PCG. Why did the authors just choose hypoalbuminemia and anemia? Are they have any correlation in the pathophysiological view.

2.     Just a correlation between two factors can't make the conclusion that these two factors have a significant association.  The number size in this manuscript was not bigger enough, so the results would be biased by some confounders.

3.     The number of adjusting parameters is not enough in the multivariable regression model, including simple linear and cox regression.

4.     To sum up, the author should do more analysis to convince us of their finding.

Author Response

The authors used the data part from their previous study. They tried to disclose that the factors of hypoalbuminemia and anemia might affect the effect of PCS and PCG on CV outcome. However,  the statistical methods used in this manuscript didn't sufficiently support its conclusion. Therefore, there were some major flaws in the manuscript.

The major comments

  1. The idea should be more precise. Some factors can affect the effect of PCS and PCG. Why did the authors just choose hypoalbuminemia and anemia? Are they have any correlation in the pathophysiological view.

As mentioned in the introduction, we choose albumin because PCS and PCG are protein bound uremic toxins, it could be assumed that the biological effects of these toxins relate to their circulating free concentrations and thus to the degree of protein binding, to mainly albumin, in the plasma.

We also mentioned in the introduction that our group previously demonstrated in a series of in vitro experiments that PBUT are also distributed within erythrocytes by active transport mechanisms involving Band 3 proteins, anion exchangers located on erythrocyte membranes. This was the pathophysiological rationale to explore the relationship with hemoglobin.

  1. Just a correlation between two factors can't make the conclusion that these two factors have a significant association. The number size in this manuscript was not bigger enough, so the results would be biased by some confounders.

In statistical terms the word correlation is used to denote association between two quantitative variables [Swinscow and Campbell, Statistics at Square One, 10th edition, BMJ books]. If the correlation is statistically significant, we can conclude that there is an association.
We agree that the strength of the association may nevertheless be weak if the correlation coefficient is small, as noted by reviewer 2. Therefore we added this to the text (page 2, section 3.1): “In the present analysis, we found a weak but significant negative correlation between serum albumin and total (T) and free (F) plasma concentrations of pCS and pCG.”

We remind the reviewer that correlation studies are univariate and confounding factors may weaken the true association. This is the reason we performed multiple linear regression analysis, where hazard ratio’s associated with one SD change in serum albumin concentration were of the same order of magnitude as other factors like age and blood pressure.

The P-values of the tested variables indicate that at least for the confounders included in our analysis, the sample size was large enough to demonstrate an independent association.

  1. The number of adjusting parameters is not enough in the multivariable regression model, including simple linear and cox regression.

We believe it is important to adjust for potential confounders that are biologically plausible and relevant to the research question.

Regarding the regression model of the factors associated with free pCS concentrations, we wanted to know whether the observed correlation between albumin and hemoglobin on one hand, and protein-bound uremic toxins on the other hand, was explained by variation in eGFR. It may be that lower albumin and hemoglobin are due to malnutrition and renal anemia associated with lower GFR and hence higher uremic toxin concentrations.

For the cox regression model, we included variables that are traditionally included in cardiovascular risk scores: age, sex, blood pressure, and diabetes. A number of additional variables were included in the initial model but were excluded by forward and/or backward statistical elimination. For your information, we include the analysis of the full set of variables below for pCS.
Results were similar for the other uremic toxins.

Abbreviation of variables

Pat_age : patient age

Pat_sex : patient gender (1=male vs. 0=female)

Pat_diabetes: diabetes status (1=yes vs. 0=no)

Pat_smoking: current smoking (1=yes vs. 0=no)

Cln_hd_sbp: systolic blood pressure

BMI: body mass index

LN-PCSf: log transformed free paracresyl sulfate

Lab_Salb: serum albumin

Lab_pre_alb: serum prealbumin

LN_CRP: log transformed C-reactive protein

lab_hdl_m: HDL cholesterol

lab_hb: hemoglobin concentration

Method = Forward Stepwise (Likelihood Ratio)

Variables in the Equation

B

SE

Wald

df

Sig.

Exp(B)

Step 1

LN_PCSf

,616

,084

54,098

1

,000

1,852

Step 2

pat_diabetes

1,013

,186

29,814

1

,000

2,754

LN_PCSf

,533

,088

36,852

1

,000

1,705

Step 3

pat_age

,035

,008

21,236

1

,000

1,036

pat_diabetes

,888

,185

22,947

1

,000

2,430

LN_PCSf

,395

,097

16,701

1

,000

1,485

Step 4

pat_age

,032

,007

19,056

1

,000

1,033

pat_diabetes

,883

,185

22,778

1

,000

2,418

LN_PCSf

,329

,097

11,586

1

,001

1,389

lab_Salb

-,103

,027

14,510

1

,000

,902

Step 5

pat_age

,027

,008

12,531

1

,000

1,027

pat_diabetes

,936

,185

25,668

1

,000

2,550

cln_hd_sbp

,017

,004

17,029

1

,000

1,018

LN_PCSf

,322

,097

11,030

1

,001

1,379

lab_Salb

-,132

,028

22,368

1

,000

,876

Step 6

pat_age

,027

,008

12,222

1

,000

1,027

pat_sex

,676

,195

12,074

1

,001

1,967

pat_diabetes

,950

,185

26,499

1

,000

2,586

cln_hd_sbp

,017

,004

16,146

1

,000

1,017

LN_PCSf

,313

,100

9,811

1

,002

1,368

lab_Salb

-,136

,028

24,057

1

,000

,872

Step 7

pat_age

,029

,008

14,485

1

,000

1,029

pat_sex

,676

,195

12,032

1

,001

1,965

pat_diabetes

1,106

,192

33,027

1

,000

3,021

cln_hd_sbp

,017

,004

15,942

1

,000

1,017

LN_PCSf

,299

,101

8,749

1

,003

1,349

lab_Salb

-,143

,028

26,352

1

,000

,867

BMI

-,054

,019

7,762

1

,005

,947

Variables not in the Equationa,b,c,d,e,f,g

Score

df

Sig.

Step 1

pat_age

27,447

1

,000

pat_sex

12,483

1

,000

pat_diabetes

32,045

1

,000

cln_hd_sbp

17,415

1

,000

lab_Salb

18,439

1

,000

lab_pre_alb

8,748

1

,003

lab_hb

8,574

1

,003

BMI

,272

1

,602

pat_smoking

11,310

1

,001

lab_hdl_m

2,092

1

,148

LN_CRP

4,869

1

,027

Step 2

pat_age

21,842

1

,000

pat_sex

13,107

1

,000

cln_hd_sbp

15,062

1

,000

lab_Salb

17,108

1

,000

lab_pre_alb

4,211

1

,040

lab_hb

4,018

1

,045

BMI

4,896

1

,027

pat_smoking

9,881

1

,002

lab_hdl_m

,172

1

,678

LN_CRP

1,642

1

,200

Step 3

pat_sex

11,135

1

,001

cln_hd_sbp

8,911

1

,003

lab_Salb

14,424

1

,000

lab_pre_alb

1,887

1

,170

lab_hb

4,718

1

,030

BMI

6,465

1

,011

pat_smoking

9,513

1

,002

lab_hdl_m

,095

1

,758

LN_CRP

,982

1

,322

Step 4

pat_sex

13,304

1

,000

cln_hd_sbp

17,060

1

,000

lab_pre_alb

,006

1

,938

lab_hb

,493

1

,483

BMI

8,814

1

,003

pat_smoking

9,508

1

,002

lab_hdl_m

,278

1

,598

LN_CRP

,067

1

,795

Step 5

pat_sex

12,528

1

,000

lab_pre_alb

,633

1

,426

lab_hb

,141

1

,707

BMI

7,823

1

,005

pat_smoking

8,841

1

,003

lab_hdl_m

1,239

1

,266

LN_CRP

,064

1

,800

Step 6

lab_pre_alb

3,363

1

,067

lab_hb

2,591

1

,107

BMI

7,747

1

,005

pat_smoking

2,650

1

,104

lab_hdl_m

,159

1

,690

LN_CRP

,502

1

,478

Step 7

lab_pre_alb

2,749

1

,097

lab_hb

2,601

1

,107

pat_smoking

2,343

1

,126

lab_hdl_m

1,449

1

,229

LN_CRP

3,284

1

,070

a. Residual Chi Square = 106,929 with 11 df Sig. = ,000

b. Residual Chi Square = 83,290 with 10 df Sig. = ,000

c. Residual Chi Square = 62,631 with 9 df Sig. = ,000

d. Residual Chi Square = 45,216 with 8 df Sig. = ,000

e. Residual Chi Square = 30,175 with 7 df Sig. = ,000

f. Residual Chi Square = 17,376 with 6 df Sig. = ,008

g. Residual Chi Square = 9,477 with 5 df Sig. = ,091

Backward stepwise elimination yielded the same final set of variables.

  1. To sum up, the author should do more analysis to convince us of their finding.

With the extra information provided, we hope the reviewer is convinced of our findings.

Reviewer 2 Report

This is a well written manuscript on the contribution of albuminemia and Hb levels to the prognostic value of Plasma P-cresyl Sulfate and P-cresyl Glucuronide for cardiovascular outcome in CKD patients.

The research design is appropriate; the methods adequately described; the results clearly presented and the conclusions supported by the results.

Author Response

This is a well written manuscript on the contribution of albuminemia and Hb levels to the prognostic value of Plasma P-cresyl Sulfate and P-cresyl Glucuronide for cardiovascular outcome in CKD patients.

The research design is appropriate; the methods adequately described; the results clearly presented and the conclusions supported by the results.

We thank the reviewer for these positive comments.

Reviewer 3 Report

In this manuscript, Verbeke et al. investigated the associations between the total and free plasma concentrations of protein-bound uremic toxins and the level of serum albumin and hemoglobin. Subsequently, they examined their predictive value for cardiovascular morbidity and mortality. In a large cohort of 523 non-dialysis CKD patients, they observed a negative correlation between albumin/hemoglobin and total and free levels of p-cresyl sulfate (pCS) and p-cresyl glucuronide (pCG) with lower levels of pCS and pCG when serum albumin and hemoglobin were higher. The study by Verbeke et al. provides important insights into the contribution of PBUTs (in particular: pCS and pCG) to the cardiovascular risk of CKD patients. Please find below my comments and questions:

1. The authors showed that both total and free levels of pCS and pCG negatively correlated with serum albumin and hemoglobin levels. Although the authors do not state anything regarding the strength of the correlation, I would consider the correlations with especially albumin to be relatively weak (as the correlation coefficient lies between -0.25 and -0.15). 

As CKD progresses, serum albumin levels decrease, and anemia becomes more prominent. The effect of albumin and hemoglobin on PBUT levels can be directly, but the decrease in albumin and hemoglobin can also reflect the severity of CKD with higher levels of PBUTs in circulation as kidney function declines. As levels of free pCS and pCG increase when albumin and hemoglobin levels decrease, does this then reflect CKD severity resulting in higher levels of toxins or does this suggest that albumin binding and erythrocyte mediated uptake of PBUTs is insufficient when albumin and hemoglobin levels decrease? In the latter case, would the ratio of free pCS/pCG versus total pCS/pCG be affected too? Would there be relatively more free pCS/pCG when albumin and hemoglobin levels are lower?

Further, the authors suggest that the binding to albumin as well as transport of PBUTs into erythrocytes can represent two complementary mechanisms that protect the vasculature from the toxic effects of the PBUTs. This would then also suggest that there are no toxic effects of uremic toxins like pCS and pCG when they are protein bound or that uremic toxin binding to albumin does not interfere in the physiological function of albumin. Also, it would suggest that there are no intracellular toxic effects of these toxins when taken up by erythrocytes and that it would not cause erythrocyte dysfunction. Can the authors comment?

2. In general, hemoglobin levels are lower in women compared to men, have the authors observed any differences in the association between hemoglobin and pCS/pCG when they specifically investigated this association in the male and female participants of this study?

3. In table 2, the authors presented the regression model of factors associated with free pCS concentrations and in the text, they stated that models for TpCS, TpCG and FpCG were similar. Also, in table 3 only the final model for log(FpCS) is included with a statement in the text that TpCS, TpCG and FpCG are similar. It would be illustrative to include these data in the manuscript as well, perhaps as a supplementary table.

Minor comments:

1. Can the authors include information on the reference values for albumin and hemoglobin in the text or figures? This would give the reader a better idea of the extent by which albumin and hemoglobin were decreased in this patient cohort. 

2. I wonder if reference 24 is the correct reference after the following statement:

“It is a major determinant of oncotic pressure, and low albumin may increase blood viscosity and increase thrombotic risk [24].” Discussion page 5, third paragraph. 

Author Response

In this manuscript, Verbeke et al. investigated the associations between the total and free plasma concentrations of protein-bound uremic toxins and the level of serum albumin and hemoglobin. Subsequently, they examined their predictive value for cardiovascular morbidity and mortality. In a large cohort of 523 non-dialysis CKD patients, they observed a negative correlation between albumin/hemoglobin and total and free levels of p-cresyl sulfate (pCS) and p-cresyl glucuronide (pCG) with lower levels of pCS and pCG when serum albumin and hemoglobin were higher. The study by Verbeke et al. provides important insights into the contribution of PBUTs (in particular: pCS and pCG) to the cardiovascular risk of CKD patients. Please find below my comments and questions:

  1. The authors showed that both total and free levels of pCS and pCG negatively correlated with serum albumin and hemoglobin levels. Although the authors do not state anything regarding the strength of the correlation, I would consider the correlations with especially albumin to be relatively weak (as the correlation coefficient lies between -0.25 and -0.15).

We agree that the strength of this univariate association is rather weak and added this to the text (page 2, section 3.1): “In the present analysis, we found a weak but significant negative correlation between serum albumin and total (T) and free (F) plasma concentrations of pCS and pCG.”

We remind the reviewer however that correlation studies are univariate and confounding factors may weaken the true association. In the multiple linear regression analysis the hazard ratio’s associated with one SD change in serum albumin concentration were of the same order of magnitude as other factors like age and blood pressure (table 3).

As CKD progresses, serum albumin levels decrease, and anemia becomes more prominent. The effect of albumin and hemoglobin on PBUT levels can be directly, but the decrease in albumin and hemoglobin can also reflect the severity of CKD with higher levels of PBUTs in circulation as kidney function declines. As levels of free pCS and pCG increase when albumin and hemoglobin levels decrease, does this then reflect CKD severity resulting in higher levels of toxins or does this suggest that albumin binding and erythrocyte mediated uptake of PBUTs is insufficient when albumin and hemoglobin levels decrease? In the latter case, would the ratio of free pCS/pCG versus total pCS/pCG be affected too? Would there be relatively more free pCS/pCG when albumin and hemoglobin levels are lower?

The fact that in multiple regression analysis, hemoglobin remains significantly negatively associated with free pCS (and the other PBUT) after including eGFR as a covariate, indicates that the potential effect of hemoglobin is at least partially independent of GFR. This was not the case for albumin.
Regarding the ratio free/total concentrations of pCS and pCG: higher albumin levels were associated with lower free concentrations of pCS and pCG, as can be expected because these are protein bound uremic toxins. This association is independent of eGFR. Ratios of free/total concentrations were not associated with hemoglobin. Because explained variance of the ratios in these regression models are small, we are careful about drawing conclusions of these findings and prefer not to speculate about underlying mechanisms. Therefor we chose to represent the relation of hemoglobin and albumin with free and total concentrations graphically in figure 1.

Further, the authors suggest that the binding to albumin as well as transport of PBUTs into erythrocytes can represent two complementary mechanisms that protect the vasculature from the toxic effects of the PBUTs. This would then also suggest that there are no toxic effects of uremic toxins like pCS and pCG when they are protein bound or that uremic toxin binding to albumin does not interfere in the physiological function of albumin. Also, it would suggest that there are no intracellular toxic effects of these toxins when taken up by erythrocytes and that it would not cause erythrocyte dysfunction. Can the authors comment?

We agree that these are only speculative mechanisms (therefore the wording ‘may’) as in an observational study one cannot infer causal relationships.
Our speculation on the potential protective effect of albumin binding is based on a previous study (reference 3 in the introduction) where the strongest association was found for free pCS in multivariate adjusted models with statistical correction for multiple comparisons. Nevertheless, we agree with the reviewer that this may only confer partial protection as in the present study also total concentrations were predictive of cardiovascular events.
Similarly, our data cannot exclude that the potential protective effect of higher number of erythrocytes may be partially offset by deleterious effects on erythrocyte function.

In summary, these compensatory mechanisms are likely insufficient to overcome the toxicity of the accumulated uremic toxins (otherwise these toxins would not be associated with outcome).
We modified the sentence in the abstract and in the conclusion as follows:
This may indicate that there are 2 pathways in blood that potentially contribute to attenuating the vasculotoxic effects of these PBUTs.

We added the word ‘try’ page 5, 3rd line: “It could be that both binding to albumin as well as transport into erythrocytes represent two complementary mechanisms that try to defend the vascular endothelium against the toxic effect of circulating uremic toxins.”

  1. In general, hemoglobin levels are lower in women compared to men, have the authors observed any differences in the association between hemoglobin and pCS/pCG when they specifically investigated this association in the male and female participants of this study?

Hemoglobin was significantly higher in men than in women (14.0 vs 13.1 g/dL) but this did not affect the association between hemoglobin and pCS/pCG as the interaction between sex and hemoglobin was not significant.

  1. In table 2, the authors presented the regression model of factors associated with free pCS concentrations and in the text, they stated that models for TpCS, TpCG and FpCG were similar. Also, in table 3 only the final model for log(FpCS) is included with a statement in the text that TpCS, TpCG and FpCG are similar. It would be illustrative to include these data in the manuscript as well, perhaps as a supplementary table.

We provided the final models for the other PBUT as supplementary tables.

Minor comments:

  1. Can the authors include information on the reference values for albumin and hemoglobin in the text or figures? This would give the reader a better idea of the extent by which albumin and hemoglobin were decreased in this patient cohort.

Reference values for albumin and hemoglobin were added to the text.

  1. I wonder if reference 24 is the correct reference after the following statement:

“It is a major determinant of oncotic pressure, and low albumin may increase blood viscosity and increase thrombotic risk [24].” Discussion page 5, third paragraph.

Thank you for noting this, the correct reference is “Hypoalbuminemia causes high blood viscosity by increasing red cell lysophosphatidylcholine.”  Jaap A. Joles, Nel Willekes-Koolschijn, and Hein A. Koomans. Kidney International, Vol. 52 (1997), pp. 761—770. The reference has been updated.

Round 2

Reviewer 1 Report

1. Such a revised manuscript still didn't offer enough analysis and evidence to convince me to believe this manuscript is ready for publication.

Reviewer 3 Report

I have no further comments, the authors have sufficiently addressed my previous questions.